# Multiple Physics Pretraining
# for Physical Surrogate Models

Michael McCabe[*,1,2]     Bruno Régaldo-Saint Blancard[1]     Liam Parker[1]     Ruben Ohana[1]

Miles Cranmer[3]     Alberto Bietti[1]     Michael Eickenberg[1]     Siavash Golkar[1]

Geraud Krawezik[1]     Francois Lanusse[1,4]     Mariel Pettee[1,5]

Tiberiu Tesileanu[1]     Kyunghyun Cho[6,7,8]     Shirley Ho[1,6,9]

The Polymathic AI Collaboration

[1] Flatiron Institute     [2] University of Colorado Boulder     [3] University of Cambridge
[4] Université Paris-Saclay, Université Paris Cité, CEA, CNRS, AIM
[5] Physics Division, Lawrence Berkeley National Laboratory
[6] New York University     [7] Prescient Design, Genentech
[8] CIFAR Fellow     [9] Princeton University

## Abstract

We introduce multiple physics pretraining (MPP), an autoregressive task-agnostic pretraining approach for physical surrogate modeling. MPP involves training large surrogate models to predict the dynamics of multiple heterogeneous physical systems simultaneously by learning features that are broadly useful across diverse physical tasks. In order to learn effectively in this setting, we introduce a shared embedding and normalization strategy that projects the fields of multiple systems into a single shared embedding space. We validate the efficacy of our approach on both pretraining and downstream tasks over a broad fluid mechanics-oriented benchmark. We show that a single MPP-pretrained transformer is able to match or outperform task-specific baselines on all pretraining sub-tasks without the need for finetuning. For downstream tasks, we demonstrate that finetuning MPP-trained models results in more accurate predictions across multiple time-steps on new physics compared to training from scratch or finetuning pretrained video foundation models.

## 1 Introduction

In recent years, the fields of natural language processing and computer vision have been revolutionized by the success of large models pretrained with task-agnostic objectives on massive, diverse datasets [1–3]. This has, in part, been driven by the use of self-supervised pretraining methods which allow models to utilize far more training data than would be accessible with supervised training [4]. These so-called "foundation models" have enabled transfer learning on entirely new scales. Despite their task-agnostic pretraining, the features they extract have been leveraged as a basis for task-specific finetuning, outperforming supervised training alone across numerous problems especially for transfer to settings that are insufficiently data-rich to train large models from scratch [5].

---

[*]Contact: mmccabe@flatironinstitute.org
   Code: https://github.com/PolymathicAI/multiple_physics_pretraining

NeurIPS 2023 AI for Science Workshop.

Deep learning for computational science has begun to see first steps in this direction. Large domain-specific pretrained models have emerged in diverse fields such as chemistry [6, 7], medicine [8, 9], astrophysics [10, 11], and climate [12] and the trend only seems to be growing as more and more models are developed for new fields both as refined versions of existing large language models and as new models trained entirely on field-specific data.

In this work, we demonstrate that similar approaches can be extended to the surrogate modeling of spatiotemporal physical systems. Spatiotemporal prediction tasks, like those found in fluids, solids, or general continuum mechanics, have attracted significant attention from the deep learning community. From direct prediction methods [13–17] to neural PDE solvers [18, 19], researchers have sought to develop fast, accurate models for physics either as faster surrogates for the partial differential equation (PDE) solvers that dominate the field or to simulate systems that cannot be exactly described or resolved by current mechanistic models and available hardware. While directly outperforming PDE solvers is difficult [20], deep learning has already begun to impact fields like atmospheric science [21–23] and cosmology [24–26], where the systems are too large or too imprecisely described to be simulated exactly.

Unfortunately, outside of a few observation-rich outliers, settings where numerical simulation is expensive or unreliable also tend to be settings where the difficulty of acquiring training data makes it impractical to train surrogates conventionally. Most deep learning-based surrogates thus far have focused on specific problems or individual families of parameterized PDEs. However, for these low-data settings, it would be valuable to have large, task-agnostic models with a broad understanding of common physical behavior to act as a foundation for finetuning.

Contributions. We introduce Multiple Physics Pretraining (MPP), a new approach for task-agnostic pretraining of physical surrogate models. Our method enables large-scale pretraining for transfer across diverse physics which we study using fluid-oriented benchmarks. Our specific contributions are:

- We develop MPP, a pretraining approach in which we embed multiple hetereogeneous physical systems into a shared embedding space and learn to autoregressively predict the dynamics of all systems simultaneously.

- We show that single transformer models pretrained with MPP are able to match or surpass modern baselines trained only on specific pretraining sub-tasks without applying task-specific finetuning to the MPP models.

- We demonstrate the transfer capabilities of models trained with MPP on systems with limited training examples (referred to as low-data systems thereafter).

- We open-source our code and provide our pretrained models at a variety of sizes for the community to experiment with on their own tasks.

## 2   Background

Notation. Let $S$ be an arbitrary physics-driven spatiotemporal dynamical systems, either described by a parameterized family of PDEs with fixed parameters, or where snapshots are gathered from observation of a unique physical phenomenon. To simplify notation, we discuss systems with a single state variable in one spatial dimension. A continuous state variable for system $S$ is represented as $u^S(x, t) : [0, L_S] \times [0, \infty) \to \mathbb{R}$. We discretize the system uniformly in space and time at resolutions $N_S$, $T_S$ respectively. A snapshot $\boldsymbol{u}_t^S \in \mathbb{R}^{N_S}$ represents the value of state variable $u^S$ at all $N_S$ spatial discretization points at time $t$. Our pretraining task is then to learn a single model $\mathcal{M}$ that can take a uniformly spaced sequence of $T_S$ snapshots $\boldsymbol{U}_t^S = [\boldsymbol{u}_{t-T_s \Delta t_S}^S, \ldots, \boldsymbol{u}_t^S]$ from system $S$ sampled from some distribution over systems and predict $\mathcal{M}(\boldsymbol{U}_t^S)$ such that $\mathcal{M}(\boldsymbol{U}_t^S) \approx \boldsymbol{u}_{t+\Delta t_S}^S$.

Autoregressive Pretraining. In vision and language, the dominant pretraining strategies include autoregressive prediction [27], masked reconstruction [2, 3], and contrastive learning [1]. In language, autoregressive generation emerged as a convenient self-supervised task. In surrogate modeling of dynamical systems, next-step prediction is often a primary goal. This

makes autoregressive pretraining a natural choice of objective for training time-dependent surrogate models.

We note that it is common to use the simulation parameters to condition the predictions of models operating on PDE-generated data [28–30]. In MPP, the model must instead implicitly infer the impact of these parameters on the dynamics from the history provided in $\boldsymbol{U}_t^S$.

Surrogate Modeling for Spatiotemporal Physical Systems.   We are primarily concerned with modeling dynamical systems varying in both time and space, where the time evolution of the system is intrinsically tied to spatial relationships amongst the state variables according to physical laws. Partial differential equations (PDEs) are one of the primary modeling tools for this setting. They are often derived from fundamental conservation laws of properties such as mass, momentum, and energy [31]. Many PDEs describe variations of the same physical laws, which is why concepts like diffusion, advection, reactivity, and connections between time and spatial gradients appear in many different PDEs. These shared underlying principles suggest we can extract features relevant to multiple physical systems.

## 3   Related Work

Foundation models.   Massive pretrained models dubbed "foundation models" [5], particularly large transformer-based architectures [32], have recently attracted significant attention. The most prevalent foundation models are pretrained language models like GPT [27, 33, 34] and BERT [3]. Emergent abilities [35] demonstrated by large language models highlight the importance of scale in manifesting higher-order capabilities absent at smaller scales. Vision has seen similar developments with the growth of masked [2, 36] and contrastive [1] pretraining. The data in this work is insufficiently diverse to call the resulting models "foundational". However, we provide the first large-scale implementation of successful multiple nonlinear physics pretraining for spatiotemporal systems.

Scientific transfer learning.   The high cost of training scientific models from scratch has led to significant exploration of transfer learning. Prior work has explored transfer learning in operator networks in such scenarios as conditional shift [37] or new domains, boundary conditions, or distributions over parameters [38–41]. However, these too need to be retrained from scratch for new differential operators in the PDE. More recently, efforts have been made to explore transfer across operators and benefits from training on multiple physical systems simultaneously. [30] in particular explores how transfer scales in this setting. However, their study is limited to steady-state linear systems with periodic boundary conditions. Other works have explored similarly restricted classes or low dimensional, low resolution systems [42, 43].

## 4   Scalable Multiple Physics Pretraining

### 4.1   Compositionality and Pretraining

Many specialized PDEs demonstrate a form of compositionality, as a range of physical phenomena can be described by core components like nonlinear advection or diffusion, but then are augmented or restricted by specialized terms representing concepts like buoyancy or system constraints. To motivate a useful pretraining procedure from this compositionality, we want to show two things:

1. Learning partially overlapping physics is beneficial for transfer learning
2. Single models can simultaneously learn many types of physics

If both of these are true, then we could train a single model which could transfer effectively to many types of physics. We start by examining the first assertion in a very simple spatiotemporal setting: constant-coefficient advection-diffusion. Let $\psi(x,t)$ be a scalar defined on a periodic spatial domain, $v$ a constant one-dimensional velocity coefficient and $\delta$

a constant diffusion coefficient, then:

$$\text{Advection:} \qquad \frac{\partial \psi}{\partial t} + \nabla \cdot (v\psi) = 0 \tag{1a}$$

$$\text{Diffusion:} \qquad \frac{\partial \psi}{\partial t} + \nabla \cdot (-\delta \nabla \psi) = 0 \tag{1b}$$

$$\text{Advection-Diffusion:} \qquad \frac{\partial \psi}{\partial t} + \nabla \cdot (v\psi - \delta \nabla \psi) = 0. \tag{1c}$$

If our first assertion is true, we would expect pretraining on the advection and diffusion terms individually could be beneficial for transfer to advection-diffusion equations.

We find that this is indeed the case. We pretrain a spatiotemporal transformer model on a large amount of trajectories (100,000 each) with uniformly sampled coefficients ($v \in [-3, 3]$, $\delta \in [10^{-3}, 1.]$) generated from the advection and diffusion equations while finetuning on restricted samples from advection-diffusion simulations. The pretrained model is able to achieve much lower error with far fewer samples (Figure 1) despite the fact that it never saw advection and diffusion occurring in the same trajectory during pretraining.

Figure 1: Finetuning a model pretrained on large amounts of advection and diffusion data outperforms models trained from scratch on advection-diffusion data across a wide range of data availability (16-100K examples).

To address question two, we must handle much larger spatial resolutions, varying scales, and heterogeneous relationships between fields. Over the rest of this section, we develop an approach for handling these challenges.

## 4.2 Architecture

**Axial Attention.** Given the success of large transformer models in other domains, we employ a scalable axial attention [44–46] transformer backbone. For a (2+1)-dimensional system with $T \times H \times W$ tokens, conventional dense attention attends over all tokens simultaneously and has cost $O((HWT)^2)$. Axial attention instead performs a series of attention operations over each axis in turn, limiting the cost to $O(H^2 + W^2 + T^2)$. In Figure 2, it can be seen that while we perform attention on each axis independently, spatial attention utilizes one set of linear projections for both the height (y) and width (x) axes.

Axial attention has been used in a number of video transformers [47, 48] due to the improved scalability in higher dimensions. While the tools used in our transformer backbone were introduced in prior work, our choice of using fully axial attention differs from ViViT which opted to only separate space and time attention. We favor scalability over maximizing accuracy and so chose the fully axial formulation. In subsequent sections we refer to this architecture as an Axial ViT (AViT).

**Field Embedding and Normalization.** Embedding multiple physical systems into a single shared representation is complicated by the fact that fields from different systems may operate on entirely different scales in terms of both magnitude and resolution. This is one of the primary challenges that must be addressed for multiple-physics pretraining.

To unify the magnitudes, we utilize reversible instance normalization [49, RevIN]. We compute the mean and standard deviation of each channel over the space-time dimensions and use them to normalize the input fields. These statistics are saved and used to denormalize the model outputs. While this approach was initially developed for time-series forecasting, the effect is similar to that reported in Subramanian et al. [30], where it was found to be beneficial to rescale the inputs to a fixed norm during training.

After rescaling, the data is projected into a shared embedding space. This is the only component with weights that are unique to each source system. Given a system $S$ with

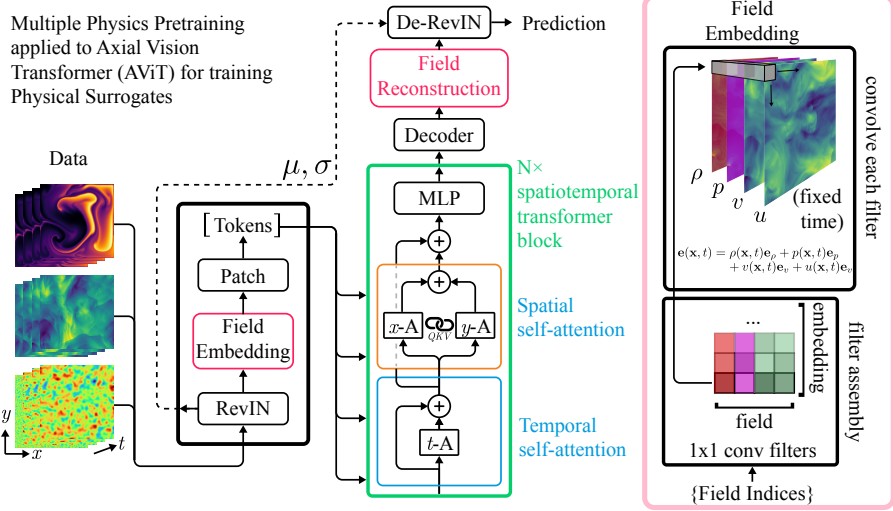

Figure 2: (Left) MPP works by individually normalizing each example using Reversible Instance Normalization (RevIN) then embedding each field individually into a shared, normalized space. A single transformer backbone can then predict the next step for multiple sets of physics. We use an AViT backbone which attends over space and time axis sequentially. Spatial attention is further split by axis, though these share linear projection weights. (Right) The embedding and reconstruction matrices are formed by subsampling a larger $1 \times 1$ convolutional filter using unique field indices passed with the input data.

state variables $u(x, t),\ v(x, t),\ p(x, t) \in \mathbb{R}$, we project each sample point or "pixel" into a space of dimension $D^{\mathrm{emb}}$:

$$\boldsymbol{e}(x, t) = u(x, t)\boldsymbol{e}_u + v(x, t)\boldsymbol{e}_v + p(x, t)\boldsymbol{e}_p \tag{2}$$

where $\boldsymbol{e}$ are embedding vectors in $\mathbb{R}^{D^{\mathrm{emb}}}$. This can be seen as a convolution with $1 \times 1$ filters where the input channels of the filter are sub-selected to correspond to the fields present within a given dataset. On the right side of Figure 2, the filter is assembled by sub-selected columns of the larger filter corresponding to the provided fields. It is important to note that this initial projection setup is amenable to fine-tuning to unseen field types. This can be achieved by adding new channels to the initial embeddings, and training them from random initialization. In our models, the shared full resolution space is converted into patched tokens by a sequence of strided convolutions separated by pointwise nonlinearities as in Touvron et al. [50].

The predictions are reconstructed from the processed tokens by reversing this process. The tokens are decoded by a sequence of transposed convolution blocks and projected onto the output fields by taking coordinate-wise inner products with reconstruction vectors $\boldsymbol{r}$:

$$u(x, t + \Delta t) = \langle \boldsymbol{e}(x, t + \Delta t), \boldsymbol{r}_u \rangle. \tag{3}$$

This can similarly be implemented as a $1 \times 1$ convolution with the output channels of the convolution filter sub-selected. The mean and standard deviation computed from the inputs are then applied to these normalized outputs to produce the final de-normalized predictions as in Kim et al. [49].

### 4.3 Balancing Objectives During Training

Task Sampling. Our pretraining procedure operates on multiple levels of sampling. The task distribution varies in system $S$, spatial resolution $N_S$, and time resolution $T_S$ and we want diverse batches that accurately capture the signal this provides. However, sampling a full batch from multiple systems at different resolutions simultaneously would be inefficient on modern hardware as it would require batch processing of differently shaped tensors. Multi-GPU training adds an additional complication as the variance in execution time due

to unbalanced workloads can lead to inefficient hardware usage. We mitigate both of these concerns with a simple randomization scheme involving gradient accumulation. Gradient accumulation utilizes multiple backward passes per synchronization step. We therefore sample a single system $S$ uniformly from $\mathcal{S}$ for each micro-batch. With $m$ micro-batches per synchronization step, we reduce the work-per-GPU variance $\sigma_{\mathcal{B}}^2$ to $\frac{1}{m}\sigma_{\mathcal{B}}^2$, significantly reducing the average lost cycles due to work discrepancies. This could likely be further reduced by an approximate packing problem solution [51], but we found the random approach was sufficient for our needs. As we employ gradient accumulation in order to increase our batch sizes, this sampling procedure incurs no additional cost.

Scaled Training Objective. The simplest approach to obtaining updates from the different tasks is to add their gradients. However, as the magnitudes of the state variables can vary significantly between systems, unweighted losses will result in the gradients from the problems with the largest scales drowning out losses on smaller scales [52]. To partially control this behavior, we train using the normalized MSE (NMSE) defined as:

$$\mathcal{L}_{\text{NMSE}} = \frac{1}{|\mathcal{B}|} \sum_{S \in \mathcal{S}} \frac{\|\mathcal{M}(\boldsymbol{U}_t^S) - \boldsymbol{u}_{t+1}^S\|_2^2}{\|\boldsymbol{u}_{t+1}^S\|_2^2 + \epsilon} \qquad (4)$$

where $\mathcal{B} \subset \mathcal{S}$ denotes the micro-batch and $\epsilon$ is a small number added for numerical stability. This does not account for the full variation in difficulty. Even if sub-task losses have similar magnitudes at the start of training, it is possible for some systems to converge quickly while other losses remain high. Nonetheless, we found that this allows our training process to produce strong results on multiple systems simultaneously.

Figure 3: Processing different physics (indicated by color) with different native resolutions incur varying wall-clock times (arrow lengths). To reduce the loss of GPU-cycles, we use gradient accumulation as a stochastic load-balancing mechanism, reducing the variance in work between all-reduce synchronizations.

## 5 Experiments

We design our experiments to probe two vital questions about the utility of MPP:

1. Can large transformer models learn the dynamics of multiple physical systems simultaneously?

2. Does MPP provide a finetuning advantage over existing spatiotemporal foundation models for new autoregressive prediction tasks?

Data. We use the full collection of two-dimensional time-dependent simulations from PDEBench [53] as our primary source for diverse pretraining data. This includes systems governed by four unique nonlinear PDEs at a variety of state variables available, resolutions, initial conditions, boundary conditions, and simulation parameters. The specific PDEs are the compressible and incompressible Navier-Stokes equations, the shallow-water equations, and a 2D Diffusion-Reaction equation. Full details on the data used can be found in Appendix B.1.

Training settings. $T^S$ is fixed at 16 for all experiments as our VideoMAE comparison in Section 5.2 was unable to scale to larger sizes without gradient checkpointing. Autoregressive training is performed only one step ahead—no longer rollouts, noise corruption, or post-processing are included for stability. Training from scratch and MPP pretraining are always performed on the AViT architecture described in section 4.2. Full training details including data splits, optimization details, and hardware are documented in Appendix C.

Table 1: NRMSE comparison between MPP-pretrained models and dedicated baselines. MPP-pretrained models learn multiple physical systems at least as well as standard baselines. Top performing within size range and overall are bolded. Dashes indicate precision not available. [†] While the PINN is much smaller, these models are fit per-example.

| Model | #Param | SWE | DiffRe2D | CNS M1.0 | CNS M0.1 |
|---|---|---|---|---|---|
| MPP-AViT-Ti | 7.6M | 0.0066 | 0.0168 | 0.0442 | 0.0312 |
| UNet | 7.7M | 0.083- | 0.84– | 0.4725 | 1.6650 |
| FNO | 466K | 0.0044 | 0.12– | 0.1685 | 0.2425 |
| PINN | 8.5K$^†$ | 0.017- | 1.6— | — | — |
| ORCA-SWIN-B | 88M | 0.0060 | 0.82– | — | — |
| MPP-AViT-B | 116M | 0.0024 | 0.0106 | 0.0281 | 0.0172 |
| MPP-AViT-S | 29M | 0.0039 | 0.0112 | 0.0319 | 0.0213 |
| MPP-AViT-L | 409M | 0.0022 | 0.0098 | 0.0208 | 0.0147 |

## 5.1 Pretraining Performance

First, we compare MPP-pretrained models to dedicated baselines from prior work across all available systems. The models are pretrained at a variety of sizes so we can begin to explore to benefits of scaling our approach. Precise model sizes can be found in Appendix C.1. Unlike the baselines which are trained on only one system and so must only learn one parameter regime, our models (denoted by MPP-AViT-*) must handle all systems and regimes without finetuning. The effect of physical parameters, forcing, and simulation parameters must be inferred from context $U_t^S$. The PINN [18], UNet [54], and FNO [13] results are sourced from Takamoto et al. [53] while the results from Shen et al. [55] with a finetuned SWIN [56] are used for ORCA. Results are reported in terms of Normalized RMSE (NRMSE, the square root of Equation 4) averaged over fields and examples, as in Takamoto et al. [29].

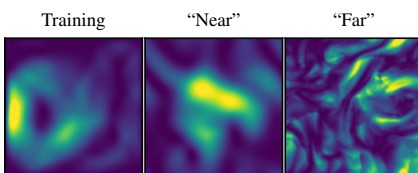

Figure 4: Kinetic energy for representative incompressible training and compressible finetuning data. The "near" compressible snapshot resembles the training snapshot while "far" displays turbulent small scales not seen in the incompressible simulation.

Our pretrained models are able achieve high-end performance on all datasets (Table 1) despite the difficulty of multi-task training [52]. In fact, there is only one case where our pretrained models do not outperform all baselines. In some cases, the improvement over the baselines is nearly an order of magnitude in NRMSE and the performance improves with scale. However, we clarify that we are not claiming these results are optimal—we can, for instance, improve upon them by finetuning our own models on specific tasks. Rather, this experiment answers affirmatively that large transformers can learn multiple sets of dynamics simultaneously. Trajectories from pretrained models are displayed in Appendix D.4.

## 5.2 Transfer to Low-data Domains

We remove all compressible fluid data from the training corpus and pretrain on the three remaining spatiotemporal systems. We evaluate transfer to two specific compressible Navier-Stokes datasets:

- "Near": $M = 0.1$, viscosity$= 10^{-2}$, Random Periodic Initial Conditions
- "Far": $M = 1.0$, viscosity$= 10^{-8}$, Turbulent Initial Conditions

Snapshots of the kinetic energy for the finetuning systems and incompressible training data are visualized in Figure 4. While quantitatively evaluating the physics gap is an unsolved problem, the names reflect both prior physical knowledge and qualitative evaluation. "Near" features a low Mach number, the dimensionless quantity that correlates with compressible

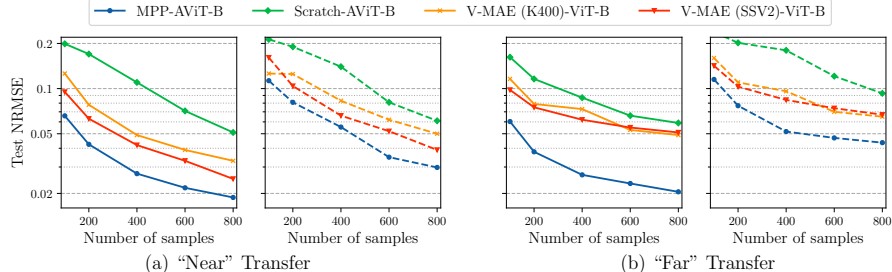

Figure 5: NRMSE for transfer learning tasks. Solid lines are one-step error. Dashed lines are averaged error over five step rollouts. The MPP model shows clear performance benefits in both cases. The more turbulent behavior of "far" seems to be difficult to learn from scratch or from video data, but pretraining on physical data leads to much stronger results.

behavior, and viscosity similar to that of the incompressible simulation. "Far" has wildly different turbulent behavior that induces small scale structure never seen during training. However, despite the similarity in physical behavior, the simulations are still quite different: the compressible and incompressible simulations in PDEBench differ in spatial and temporal resolution, initial condition distribution, boundary conditions, viscosity, and velocity range in addition to the difference in compressibility. We use these sets to compare the finetuning performance of MPP, training from scratch, and an existing pretrained spatiotemporal transformer, VideoMAE [36] pretrained on both K400 [57] and SSV2 [58] datasets.

Figure 5 shows that the MPP models outperform Video-MAE and training from scratch by a large margin in the low-data regime. Numerical results are listed in Appendix C. VideoMAE displays surprisingly strong finetuning performance given that the pretraining data is conventional video, but it is unable to match the much lower memory (Table 2) MPP-AViT-B in either setting. Predictably, both pretraining approaches are less accurate in the long-run on the turbulent "far" dataset. However, in the short-term the physical pretraining seems to provide an even larger advantage in this regime compared to the far smoother "near" data. Rollout visualizations are included in Appendix D.5.

Table 2: Memory usage during finetuning on $16 \times 3 \times 512 \times 512$ inputs for batch size 1 using mixed precision.

| Model | Max Memory |
| --- | --- |
| VideoMAE | 79.3 GB |
| AViT-B | 24.7 GB |
| AViT-Ti | 6.7 GB |
| AViT-S | 11.5 GB |
| AViT-L | 59.7 GB |

## 6  Conclusion

Limitations and Future Work.  Creating a true foundation model for fluids, continuum mechanics, or general physics requires significantly more data diversity capturing far more behavior at resolutions that are practically useful to researchers in these fields than what is included in this paper. Additionally, the architecture used in our current work assumes uniformly gridded data. Training a foundation model that can be extended to engineering-grade problems requires the ability to handle highly non-uniform grids and arbitrary geometries. Nonetheless, this work addressed important roadblocks in the development of foundation models for these fields.

The worlds of science and engineering are filled with complex phenomena that could tremendously benefit from fast surrogates, but that are lacking sufficient data for training those surrogates. Our approach, Multiple Physics Pretraining, offers new opportunities for training highly transferable models for use in these settings. We demonstrated that transformers are able to be finetuned effectively when trained on partially overlapping physics. This suggests value in large pretrained models trained on diverse physics. Our experiments showed transformers pretrained with MPP learn multiple sets of physics competitively with many dedicated approaches and that this knowledge transfers even across significant physical gaps. As physical datasets for machine learning mature, this capability paves the way for the development of true foundation models for spatiotemporal physics.

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

## A    Experiment on Broader Usage of Pretrained Representations

One of the fascinating aspects of large pretrained models is the utility of their learned features for entirely new types of prediction problems. We explore this behavior with the inverse problem of parameter estimation for two parameters:

Forcing Identification for Incompressible Navier-Stokes We attempt to identify the constant forcing term used in the incompressible Navier-Stokes simulation from an input trajectory $U_t^S$. We divide the validation set from pretraining, taking 1,000 trajectories as the new training set and using the rest for validation. Results are reported on the original test set.

Buoyancy for Incompressible Navier-Stokes For this, we turn to an additional fluid mechanics benchmark, PDEArena [28]. This benchmark includes an incompressible Navier-Stokes simulation with variable buoyancy. Since this set was not used during training, we take 1,000 randomly sampled trajectories for train, 100 for validation, and a further 1,000 for testing.

We see mixed results (Table 3). Pretraining reduces the error in the forcing task by nearly half, but largely fails to outperform the optimal constant prediction in the buoyancy task. Prior work [59] outperformed this constant prediction on buoyancy through Lie-transformation based contrastive pretraining using a convolutional architecture, so the task does appear to be possible. Since the AViT trained from scratch also fails to outperform a mean prediction, this is not a failure of MPP specif-

Table 3: RMSE for inverse problem tasks. Error from constant prediction included for context.

| Training | Forcing | Buoyancy |
|---|---|---|
| MPP | $0.20^{\pm.008}$ | $0.78^{\pm.006}$ |
| Scratch | $0.43^{\pm.012}$ | $0.77^{\pm.005}$ |
| Best Constant | $1.00^{\pm.000}$ | $0.77^{\pm.000}$ |

ically, but it is an interesting observation. It is plausible that the use of the same attention weights in both spatial dimensions makes it difficult to disentangle directional magnitudes for scalar prediction. However, at this stage, it appears the MPP-pretrained model has significant advantages on the dense prediction task that strongly resembles the pretraining task, but no visible advantages for scalar prediction.

## B    Data Details

### B.1    PDEBench

To train and evaluate our models, we use the publicly available PDEBench dataset[2] [53]. We summarize the data included in this section. This dataset comprises a suite of time dependent and time independent simulations based on common PDE systems, generated with varying parameters, initial conditions, and boundary conditions. Specifically, PDEBench uses a discretized ground-truth solver with high precision to evolve the vector-valued solution to a given PDE at one time step to the solution at one time step later. When compiled across time steps, the vector-valued solutions take the form $x \in \mathbb{R}^{T \times C \times H \times W}$, where $T$ denotes the total number of times steps, $H$ and $W$ denote the spatial height and width of the simulation grid and $C$ denotes the parameter space representing the velocity ($v_x$ and $v_y$), pressure ($p$) and density ($\rho$) fields, such that $C = 4$. For our study, we focus on the 2D fluid dynamics simulations in PDEBench. These are outlined loosely below; for more details, we refer the reader to Takamoto et al. [53]:

Compressible Navier-Stokes: These equations are used to model the pressure and velocity of both laminar and turbulent Newtonian fluids, and are applied to many real-world problems, from aerodynamics to interstellar gas dynamics. In the regime in which the density of the

---

[2]https://github.com/pdebench/PDEBench

fluid can change due to pressure variation, the equations can be expressed:

$$\partial_t \rho + \nabla \cdot (\rho \mathbf{v}) = 0, \tag{5}$$

$$\rho \left( \partial_t \mathbf{v} + \mathbf{v} \cdot \nabla \mathbf{v} \right) = -\nabla p + \eta \nabla^2 \mathbf{v} + (\zeta + \eta/3)\nabla(\nabla \cdot \mathbf{v}) \tag{6}$$

$$\partial_t(\epsilon + \rho v^2/2) + \nabla \cdot \left[ (p + \epsilon + pv^2/2)\mathbf{v} - \mathbf{v} \cdot \boldsymbol{\sigma}' \right] = \mathbf{0}, \tag{7}$$

where $\rho$ is the fluid density, $\mathbf{v}$ is the fluid velocity, $p$ is the fluid pressure, $\epsilon$ is the internal energy, $\boldsymbol{\sigma}'$ is the viscous stress tensor, $\eta$ is the shear viscosity, and $\zeta$ is the bulk viscosity. For our transfer experiments, we use the following two sets of data in particular:

1. A set of 1,000 trajectories on a $H \times W = 512 \times 512$ regular grid over $T = 100$ time steps (where the separation between steps is $\Delta t = 0.005$). Additionally, $(M, \eta, \zeta) = (1.0, 10^{-8}, 10^{-8})$, where $M, \eta, \zeta$ denote the Mach number, the shear viscosity, and the bulk viscosity, respectively. The velocity field is initialized with a turbulent field, while the inital pressure and density fields are taken to be uniform.

2. A set of 10,000 trajectories on a $H \times W = 128 \times 128$ regular grid with $(M, \eta, \zeta) = (0.1, 0.01, 0.01)$. The time steps and initializations are as above.

Incompressible NS: In the incompressible regime, which typically occurs in fluids with low Mach numbers (as it rules out density and pressure waves like sound or shock waves), the Navier-Stokes equations simplify to:

$$\nabla \cdot \mathbf{v} = 0, \tag{8}$$

$$\rho \left( \partial_t \mathbf{v} + \mathbf{v} \cdot \nabla \mathbf{v} \right) = -\nabla p + \eta \nabla^2 \mathbf{v} + \mathbf{f}, \tag{9}$$

where $\mathbf{v}$ is the velocity, $\rho$ is the density, $p$ is the pressure, $\eta$ is the viscosity, and $\mathbf{f}$ is the external force. The simulation in PDE bench is augmented by an immersed tracer that is transported by the velocity field:

$$\partial_t \rho_{smoke} = -\mathbf{v} \cdot \nabla \rho_{smoke} \tag{10}$$

These equations are typically used to model a variety of hydrodynamics systems such as weather. This data is produced at resolution $512 \times 512$ with time step of .0005. The dataset contains a total of 1000 trajectories with 1000 time steps each.

Shallow water: In the event that the horizontal length scale of the fluid is significantly greater than the vertical length scale, the incompressible Navier-Stokes equations can be depth-integrated to derive the shallow water equations. These describe flow below a pressure surface in a fluid, and are given by

$$\partial_t h + \nabla \cdot (h\mathbf{v}) = 0, \tag{11}$$

$$\partial_t(h\mathbf{v}) + \nabla \cdot \left( \frac{1}{2} h\mathbf{v}^2 + \frac{1}{2} g_r h^2 \right) = -g_r h \nabla b, \tag{12}$$

where $h$ is the water depth, $\mathbf{v}$ is the velocity, $b$ is the bathymetry, and $g_r$ is the reduced gravity. For our data, we use 1,000 trajectories on a $H \times W = 128 \times 128$ regular grid over $T = 100$ time steps. The specific simulation used is a 2D radial dam break scenario, where the water height is initialized as a circular bump in the center of the domain with a uniformly randomly sampled radius.

Diffusion-Reaction: The Diffusion-Reaction equations arise in systems with many interacting components and can be represented in the general form

$$\partial_t \mathbf{u} = \mathbf{D}\nabla^2 \mathbf{u} + \mathbf{R}(\mathbf{u}), \tag{13}$$

where $\mathbf{u}$ is a vector of concentration variables, $\mathbf{D}$ is a diagonal matrix of diffusion coefficients, and $\mathbf{R}$ describes all local reaction kinetics. The most common application of diffusion-reaction equations is in chemical reactions, however they can also be used to describe a variety of dynamical processes. For our data, we use 1,000 trajectories on a $H \times W = 128 \times 128$ regular grid over $T = 100$ time steps. The reaction functions for the activator and inhibitor are defined by the Fitzhugh-Nagumo equation [60], and their diffusion coefficients are $D_u = 1 \times 10^{-3}$ and $D_v = 5 \times 10^{-3}$ respectively. The initial conditions are generated as standard Gaussian random noise.

## B.2 PDEArena

In addition to the 2D Incompressible Navier-Stokes data incorporated from PDEBench, we also include 2D Incompressible Navier-Stokes data from PDEArena [28]. This includes a set of 5,200 training trajectories (and 1,300 validation and test trajectories each) on a $H \times W = 128 \times 128$ regular grid from which we take $T = 16$ timesteps for prediction. As with the PDEBench simulations, the PDEArena simulations include a viscosity parameters of $\nu = 0.01$ and Dirichlet boundary conditions, however they also include a buoyancy term $f \in [0.2, 0.5]$ in the $y$ direction.

## C  Experiment Details

### C.1  Model Configurations

The following architectural decisions were used across all AViT models trained in this paper:

- Pre/Post Norm: Pre-norm [61]
- Normalization Type: Instance Normalization [62]
- Activations: GeLU [63]
- QK Norm: Yes [64]
- Patching: hMLP [50]
- Decoder: Transposed hMLP (this is equivalent to the transposed convolutions mentioned in the main text).
- Causal Masking: False - We only evaluate the loss on the $T + 1$ prediction.

Furthermore, we examine the performance of our models on the aforementioned PDE systems when the size of the model is scaled. Vision transformers have a variety of parameters that control the model's size, including the number of processor blocks, the dimensionality of patch embeddings and self-attention, the dimensionality of Multi-Layer Perceptron (MLP) blocks, the number of attention heads, and the patch size applied on the input tensors. In previous studies on language [65–67] and vision [68], it has generally been noted that model performance is typically only weakly dependent on shape parameters, and instead depends largely on non-embedding parameter count given a fixed compute budget and dataset size. As such, we follow the general scaled architectures set forth by Zhai et al. [68] for vision, and scale all aspects of the model shapes simultaneously to select a variety of model sizes for testing. These are detailed in 4.

**Position Biases and Boundaries.**  While in most cases, we would like the model to infer boundary conditions from the provided history, we make an exception to this policy for periodic boundaries as they change the continuity of the domain. Transformers are inherently permutation equivariant, and it is essential to include position biases so that the model can learn locality.

With a slight modification, we can use our position biases to capture the change in locality imposed by periodic boundaries. T5-style [69] relative position encodings (RPE) utilize a lookup table to access learned embeddings corresponding to ranges of "relative distance". For periodic boundary conditions, we modify the relative distance computation to account for neighbors across the periodic boundary. We find that this minor change enables generalization to periodic boundary conditions whether or not they are included in the training data.

**Software.**  All model development and training in this paper is performed using PyTorch 2.0 [70].

**Hardware.**  All training for both pretraining and finetuning is done using Distributed Data Parallel (DDP) across 8 Nvidia H100-80GB GPUs.

Table 4: Details of the various model architectures and scales explored.

| Model | Embed Dim. | MLP Dim. | # Heads | # Blocks | Patch Size | # Params |
|---|---|---|---|---|---|---|
| AViT-Ti | 192 | 768 | 3 | 12 | [16, 16] | 7.6M |
| AViT-S | 384 | 1536 | 6 | 12 | [16, 16] | 29M |
| AViT-B | 768 | 3072 | 12 | 12 | [16, 16] | 116M |
| AViT-L | 1024 | 4096 | 16 | 24 | [16, 16] | 409M |

## C.2 Exp 1: Pretraining Performance

For both MPP and scratch models, we train using the following settings:

- Training Duration: 200K steps
- Train/Val/Test: .8/.1/.1 split per dataset on the trajectory level.
- Task sampling: Uniformly sample task, then uniformly sample trajectory from task without replacement. We treat every 400 model updates (1 model update=5 micro-batches) as an "epoch" and reset the task pool.
- Micro-batch size: 8
- Accumulation Steps: 5
- Optimizer: Adan [71]
- Weight Decay: 1E-3
- Drop Path: 0.1
- Base LR: DAdaptation [72]
- LR Schedule: Cosine decay
- Gradient clipping: 1.0

Note, we use the automated learning selection strategy DAdaptation during pretraining runs in large part to avoid excessive hyperparameter tuning of our own models. In finetuning experiments, comparison models are tuned manually following the recommended settings from the model publishers to avoid differences being due to compatibility with the parameter-free method.

Data   For pretraining, we use all PDEBench datasets. These are described in Section B.1. In particular, we use the compressible and incompressible Navier-Stokes, Diffusion-Reaction 2D, and Shallow Water data.

## C.3 Experiment 2: Transfer to Low-Data Domains

In this experiment, we compare the transferability of our MPP-Pretrained models to general-purposes pretrained video masked autoencoders [VideoMAE; 36] for frame prediction on video-like PDEBench data [53].

For MPP and training from scatch, we use the following settings:

- Training Duration: 500 epochs
- Train/Val/Test: X/.1/.1 split per dataset on the trajectory level. Note that X is due to the fact that we test varying amounts of training data. These are subsampled from the training split of 80%.
- Batch size: 8
- Accumulation Steps: 1 (No accumulation)
- Optimizer: Adan [71]
- Weight Decay: 1E-3
- Drop Path: 0.1

- Base LR: DAdaptation [72]
- LR Schedule: Cosine decay
- Gradient clipping: 1.0

**Data**  We study transferability of VideoMAE models for spatiotemporal prediction on video-like scientific data.

AViT Models are pretrained on datasets generated from three PDEs: Incompressible Navier-Stokes, Shallow Water, and Diffusion Reaction 2D.

We focus on transfer to the two datasets "Near" and "Far" (see Sect. 5.2) of fluid dynamics simulations taken from the PDEBench dataset [53]. These simulations solve the compressible Navier-Stokes equations in a 2D geometry with periodic boundary conditions (see Appendix B.1 for additional details).

### C.3.1  VideoMAE Settings

While VideoMAE does utilize spatiotemporal information, it was developed for a different setting, so we fully document all details of our adaptation of it here both for reproducibility and fairness in our comparison.

VideoMAE models are video transformers that were proven to be efficient data-learners for self-supervised video pretraining [36]. They rely on an asymmetric encoder-decoder architecture building on a vanilla ViT backbone with joint space-time attention. VideoMAE models are pretrained by learning to reconstruct masked videos using a random tube-masking strategy with a extremely high masking ratio ($\sim 90\,\%$).

We make use of two publicly available models, hereafter called VideoMAE-K400 and VideoMAE-SSV2, that were pretrained on Kinetics-400 dataset [K400; 57] and Something-Something V2 dataset [SSV2; 58], respectively. Both datasets are made of short videos (typically $\leq 10\,s$ long) of human-object or human-human interactions. VideoMAE-K400 (respectively, VideoMAE-SSV2) was pretrained on $\sim 240$k ($\sim 170$k) videos. We focus on the models that build on a ViT-base backbone, so that their size (in terms of number of trainable parameters) remains comparable to that of MPP-AViT-B. After adaptation of the input and output linear layers as described below, the number of trainable parameters of these models reaches $\sim 95$ M.

**Number of channels.**  Same as the original pretraining procedure, the input data $x \in \mathbb{R}^{C \times T \times H \times W}$ is divided into non-overlapping joint space-time cubes of size $2 \times 16 \times 16$. These are embedded through a Conv3d layer, resulting in $\frac{T}{2} \times \frac{H}{16} \times \frac{W}{16}$ tokens. Since our PDEBench data has $C = 4$ channels instead of 3 for the RGB videos from the pretraining set, we had to adapt the number of input channels of this Conv3d layer accordingly. The weights of this new layer were defined using a (rescaled) repetition of the pretrained weights from the original layer. Similarly, the output number of features of the final linear projection layer of the model had to be adapted to $C = 4$ channels. The weights and biases of this layer were extended by consistently repeating the original pretrained weights and biases.

**Positional encoding.**  The number of tokens resulting from our PDEBench data did not match the number of tokens resulting from the pretraining datasets. Consequently, we also had to adapt the pretraining positional encoding. We chose to interpolate accordingly the original 1D sine/cosine positional encoding [32] using a trilinear interpolation after having reshaped the token index axis onto a 3D grid.

### C.3.2  Video MAE Finetuning Procedure

We describe the finetuning procedure of the pretrained VideoMAE models for frame prediction. Frame prediction consists in predicting the next $T_p$ frames of a video given a context of $T_c$ frames. Since the pretrained models manipulates space-time cubes of size 2 in time, we naturally choose $T_p = 2$. The context size is taken to be $T_c = 16$ for consistency with MPP-AViT models. We finetune the pretrained models for frame prediction by adapting

Table 5: Effective learning rate for the finetuning of VideoMAE.

|  | "Near" | "Far" |
|---|---|---|
| VideoMAE (K400) | 0.00039 | 0.00198 |
| VideoMAE (SSV2) | 0.00186 | 0.00150 |

the self-supervised training strategy in order to reconstruct the last $T_p$ frames of a masked video of $T = T_c + T_p$ frames.

**Masking strategy.** For frame prediction, instead of the random tube-masking strategy, we simply mask the last $T_p$ frames of the input data.

**Loss.** We finetune our models by minimizing a NMSE loss. In this context, denoting by $x, y \in \mathbb{R}^{C \times T_p \times H \times W}$ the output of our model and the target (masked frames), respectively, the NMSE loss is defined by $\mathcal{L}(x, y) = \sum_{c=1}^{C} \sum_{t=1}^{T_p} \|x_{c,t} - y_{c,t}\|_2^2 / \|y_{c,t}\|_2^2$.

**Normalization of the data.** Each set of PDEBench simulations is globally and channel-wise rescaled so that pixel values all fit in $[0, 1]$. Additionally, we normalize channel-wise the targets $y \in \mathbb{R}^{C \times T_p \times H \times W}$ by subtracting the global mean of the corresponding context frames and then dividing by their global standard deviation.

**Optimization.** We finetune the pretrained models over 500 epochs and a (total) batch size of 8 using AdamW optimizer [73]. Except for the learning rate, the remaining optimization hyperparameters are chosen to be consistent with those used in the finetuning experiments of [36] (Table 10). In particular, we choose a weight decay $\lambda = 0.05$, $(\beta_1, \beta_2) = (0.9, 0.999)$, a cosine learning rate decay scheduler with 5 warmup epochs, a drop path rate of 0.1, and a layer-wise learning rate decay parametrized by 0.75. In this setting, the learning rate is adjusted by performing a hyperparameter search monitored with WandB [74]. We report the resulting optimal values per pretrained model and dataset in Table 5.

### C.4 Exp 3: Broader Usage of Pretrained Representations

For MPP and training from scatch, we use the following settings:

- Training Duration: 500 epochs
- Train/Val/Test: 1000/100/1000 taken from original validation set or randomly depending on whether data was used for training.
- Batch size: 24
- Accumulation Steps: 1 (No accumulation)
- Optimizer: Adan [71]
- Weight Decay: 1E-3
- Drop Path: 0.1
- Base LR: DAdaptation [72]
- LR Schedule: Cosine decay
- Gradient clipping: 1.0

## D Additional and Extended Results

### D.1 Position Bias Evaluation

We isolate the impact of position biases on our multi-task training objectives by constructing an experiment that isolates their influence. Recall the advection equation from Equation 1:

$$\frac{\partial \psi}{\partial t} + \nabla \cdot (v\psi) = 0 \tag{14}$$

Table 6: Validation NRMSE for position bias comparison. Compares training performance on data that differs only in boundary conditions.

| Training | Periodic | Absorbing |
|---|---|---|
| Periodic Baseline | 0.032 | — |
| Absorbing Baseline | — | 0.295 |
| Combined | | |
|   Standard RPE | 0.188 | 0.189 |
|   Periodic-Adjusted RPE | 0.081 | 0.143 |

Table 7: Per dataset NRMSE comparison for $M = 0.1$ Compressible Navier-Stokes data. R/T denote "random" and "turbulent" initial conditions from PDEBench. $\eta = \zeta$ are the bulk and sheer viscosity.

| Model | R-$\eta = 10^{-8}$ | R-$\eta = 10^{-2}$ | R-$\eta = 10^{-1}$ | T-$\eta = 10^{-8}$ |
|---|---|---|---|---|
| MPP-AViT-Ti | 0.0493 | 0.0274 | 0.0116 | 0.0339 |
| UNet | 0.66– | 0.71– | 5.1— | 0.19– |
| FNO | 0.28– | 0.17– | 0.36– | 0.16– |
| MPP-AViT-S | 0.0335 | 0.0176 | 0.0071 | 0.0217 |
| MPP-AViT-B | 0.0286 | 0.0162 | 0.0078 | 0.0169 |
| MPP-AViT-L | 0.0234 | 0.0145 | 0.0099 | 0.0136 |

We will define two sets of physics. In both cases, the function is defined on the 1D domain $x \in [0,1]$. We sample $v \sim Unif(-1,1)$ and use initial conditions sampled from the set of circular Gaussians with variances sampled from $Unif(1/160, 1/5)$ and means sampled from $Unif(.25, .75)$. The two systems vary only in the choice of boundary conditions. The first uses periodic boundary conditions, implying $\phi(0) = \phi(1)$. The second uses absorbing boundary conditions in which waves are not reflected back into the solution space. The restricted functional form allows us to implement this exactly by extending the domain and solving the periodic equations such that the constant velocity implies the waves exiting the solution space never return.

In this experiment, we first train models (AViT-Ti with 1D patches) on each system individually using 10,000 examples each for 100 epochs to get a sense of the baseline performance. We then train models with and without our modified position biases on the two systems jointly (20,000 examples) to evaluate the impact of our change.

Table 6 shows that our modified position biases are more effective at training in the joint setting. Both RPE schemes are able to improve on absorbing boundary with the additional data. Standard RPE on the other hand struggles to learn the periodic baseline. Our Periodic-adjusted variant is much more effective at learning the periodic data, though it does not outperform the baseline.

It is interesting to note how large the effect of boundary conditions is on this problem. The model trained on only periodic condition reaches nearly an order of magnitude higher precision. While absorbing boundaries are complicated for numerical solvers, it seems as though attention should be able to simply not attend to waves passing out of the domain. The interaction of boundary conditions with attention therefore seems to be an important direction for future study.

### D.2   Exp1: CNS Expanded Results

Here we break out the Compressible Navier-Stokes (CNS) results from Table 1. Table 1 shows the comparison between our pretrained models and task-specific baselines; however, due to space limitations the CNS was aggregated by mach number in the main text, so we share the full CNS results here. M0.1 can be seen in Table 7. M1.0 can be seen in Table 8. Note that while it is conventional to describe these simulations in terms of dimensionless

Table 8: Per dataset NRMSE comparison for $M = 1.0$ Compressible Navier-Stokes simulations. R/T denote "random" and "turbulent" initial conditions from PDEBench. $\eta = \zeta$ are the bulk and sheer viscocity.

| Model | R-$\eta = 10^{-8}$ | R-$\eta = 10^{-2}$ | R-$\eta = 10^{-1}$ | T-$\eta = 10^{-8}$ |
|---|---|---|---|---|
| MPP-AViT-Ti | 0.0615 | 0.0327 | 0.0171 | 0.0594 |
| UNet | 0.47– | 0.36– | 0.92– | 0.14– |
| FNO | 0.35– | 0.096- | 0.098– | 0.13– |
| MPP-AViT-S | 0.0451 | 0.0223 | 0.0108 | 0.0425 |
| MPP-AViT-B | 0.0386 | 0.0195 | 0.0119 | 0.0365 |
| MPP-AViT-L | 0.0314 | 0.0171 | 0.0132 | 0.0282 |

Table 9: Test NRMSE for "Near" Compressible Navier-Stokes M0.1, $\eta = .01$.

| Model | # Training Samples (NRMSE $\times 10^{-1}$) | | | | | | | | | |
|---|---|---|---|---|---|---|---|---|---|---|
| | 100 | | 200 | | 400 | | 600 | | 800 | |
| | T+1 | T+5 | T+1 | T+5 | T+1 | T+5 | T+1 | T+5 | T+1 | T+5 |
| VideoMAE (K400) | 1.26 | 1.98 | 0.78 | 1.25 | 0.49 | 0.83 | 0.39 | 0.62 | 0.33 | 0.50 |
| VideoMAE (SSV2) | 0.95 | 1.61 | 0.63 | 1.04 | 0.42 | 0.66 | 0.33 | 0.52 | 0.25 | 0.39 |
| MPP-AViT-B | 0.66 | 1.13 | 0.42 | 0.81 | 0.27 | 0.55 | 0.22 | 0.35 | 0.19 | 0.30 |

numbers like the Reynolds number, these simulations are performed at relatively low resolution, so it is likely they incur significant numerical diffusion. Thus we report the results in terms of the nominal diffusion coefficients without making claims about the Reynolds numbers of the simulation.

In examining the full CNS data, one interesting result jumps out - the most viscous systems $\eta = .1$ seem to perform relatively worse with scale. For both subsets, S was the top performing model at the highest viscosity. All other viscosities seem to benefit from scale. This does seem to have a limit, however, as Ti again loses performance. It is also important to remember that these results occur during multi-task training, so they cannot be directly interpreted in the single-task setting.

### D.3   Exp2: Numerical Results

We provide numerical results corresponding to Figure 5 in Tables 9 and 10. We refer to Sect. 5.2 for discussion.

### D.4   Pretraining Trajectories

Here we show example trajectories from pretrained models. Videos are included in the attached supplementary material. After pretraining, we find that the model initially produces strong predictions, but patch artifacts creep in over time.

### D.5   Finetuning Trajectories

After finetuning, we find that the patch-based instability mostly disappears. Again, videos displaying longer trajectories are available in the supplementary material.

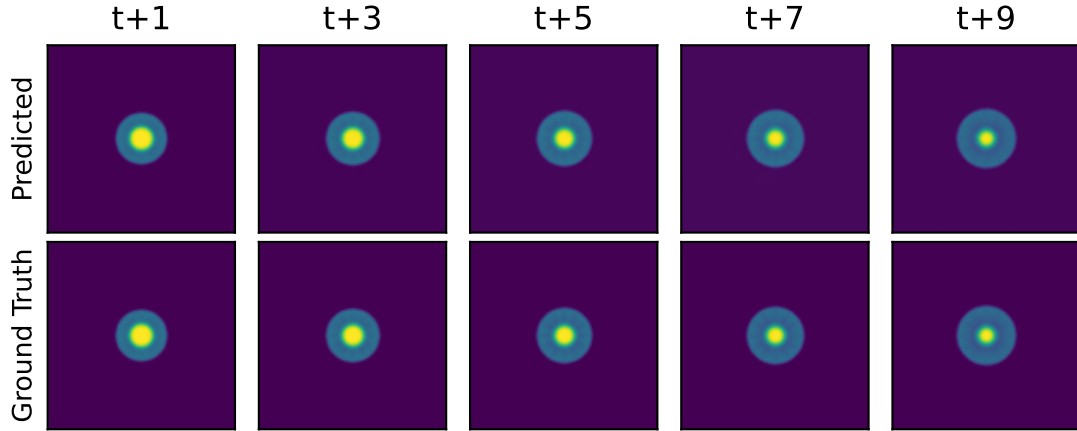

Dynamics: swe, Field h

Figure 6: Pretraining trajectory.

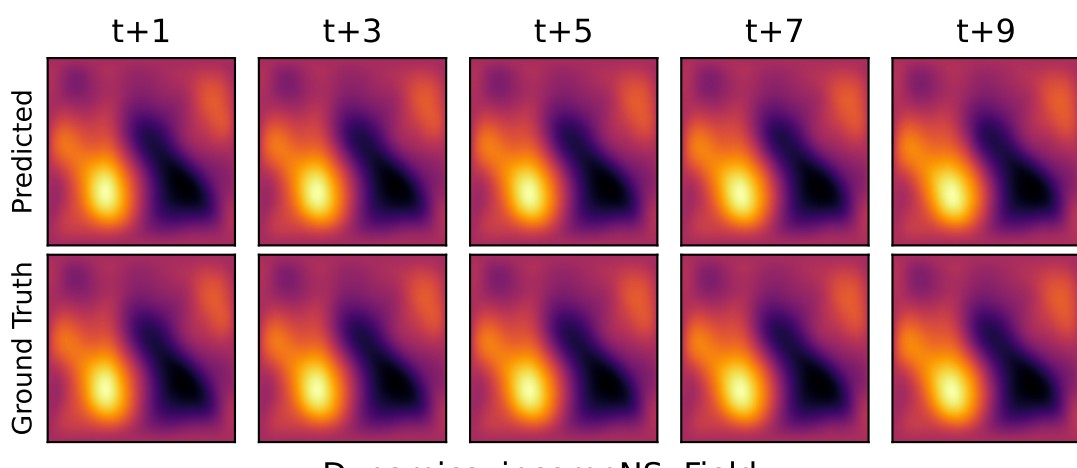

Dynamics: incompNS, Field vx

Figure 7: Pretraining trajectory.

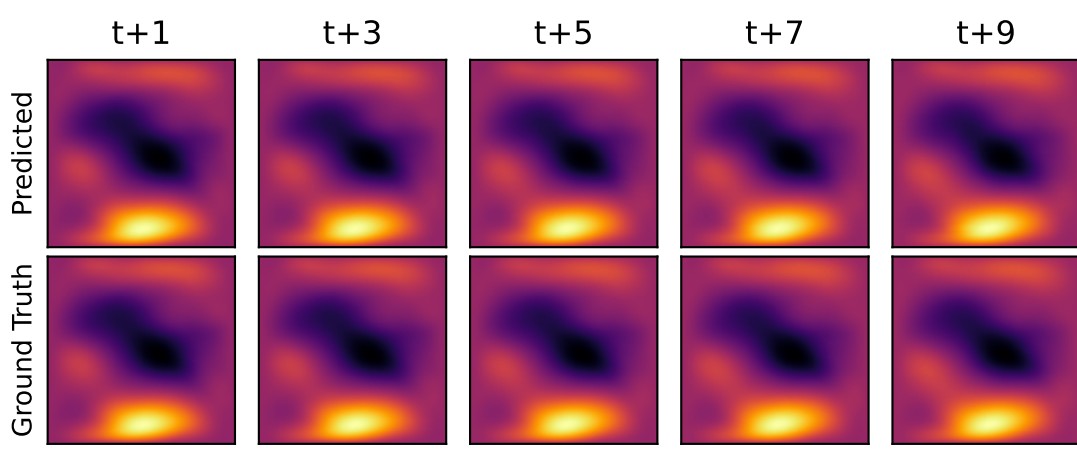

Dynamics: incompNS, Field vy

Table 10: Test NRMSE for "Far" Compressible Navier-Stokes

| Model | # Training Samples (NRMSE $\times 10^{-1}$) | | | | | | | | | |
| | 100 | | 200 | | 400 | | 600 | | 800 | |
| | T+1 | T+5 | T+1 | T+5 | T+1 | T+5 | T+1 | T+5 | T+1 | T+5 |
|---|---|---|---|---|---|---|---|---|---|---|
| VideoMAE (K400) | 1.16 | 1.60 | 0.79 | 1.10 | 0.73 | 0.96 | 0.53 | 0.70 | 0.49 | 0.65 |
| VideoMAE (SSV2) | 0.98 | 1.42 | 0.75 | 1.03 | 0.62 | 0.84 | 0.55 | 0.74 | 0.51 | 0.67 |
| MPP-AViT-B | 0.60 | 1.15 | 0.37 | 0.77 | 0.27 | 0.66 | .32 | 0.63 | 0.24 | 0.48 |

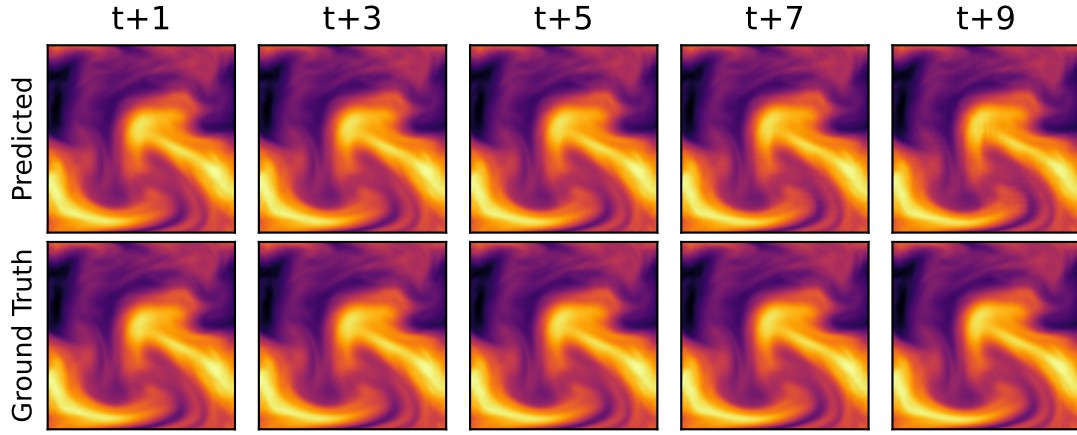

Dynamics: incompNS, Field particles

Figure 8: Pretraining trajectory.

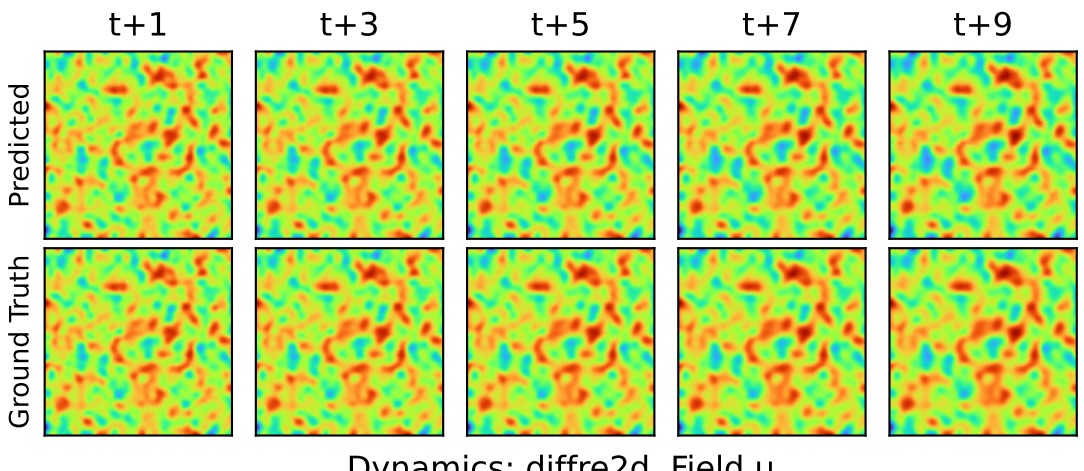

Dynamics: diffre2d, Field u

Figure 9: Pretraining trajectory.

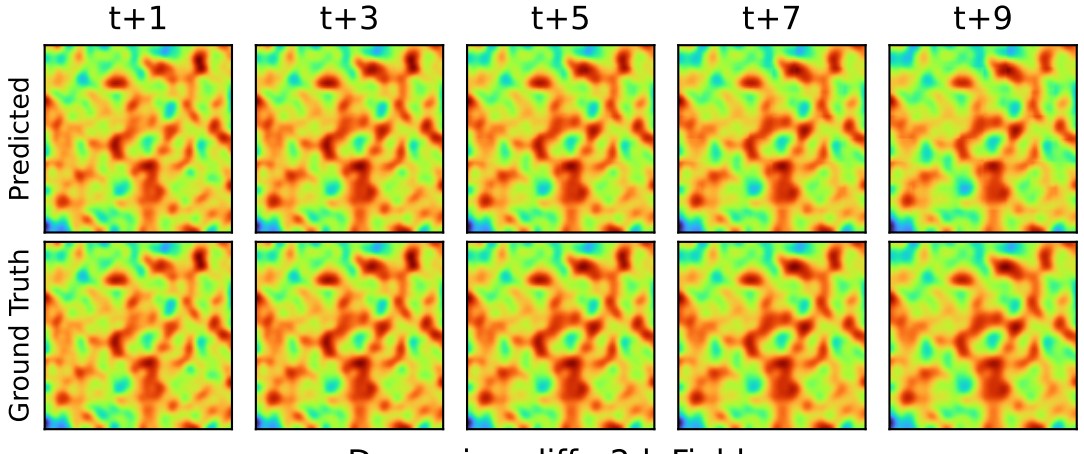

Figure 10: Pretraining trajectory.

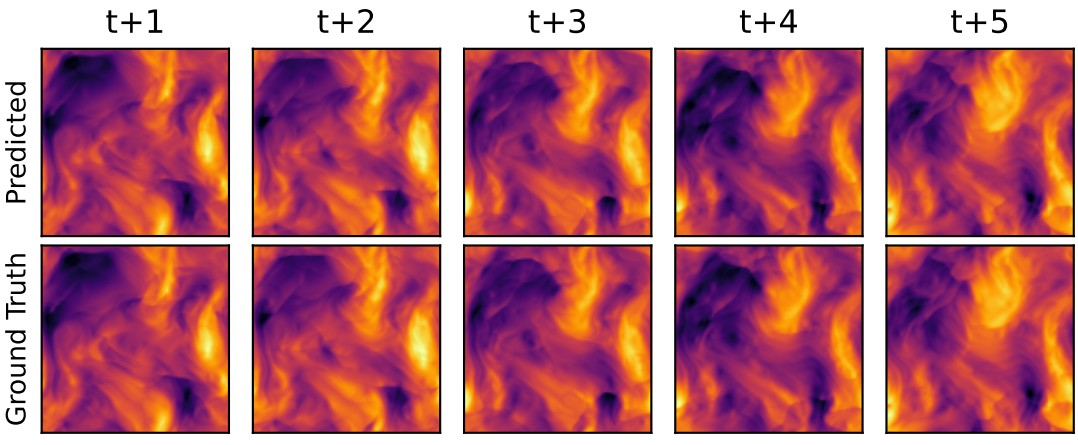

Figure 11: Finetuning trajectory.

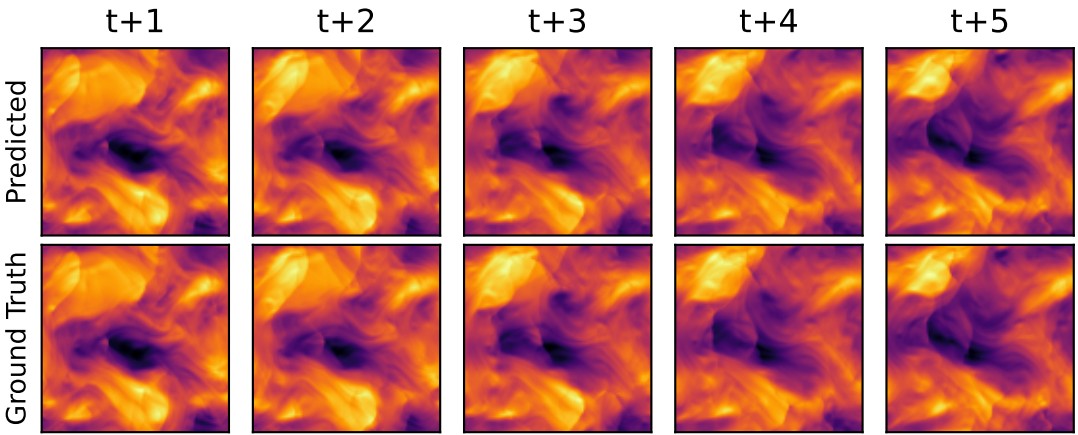

Figure 12: Finetuning trajectory.

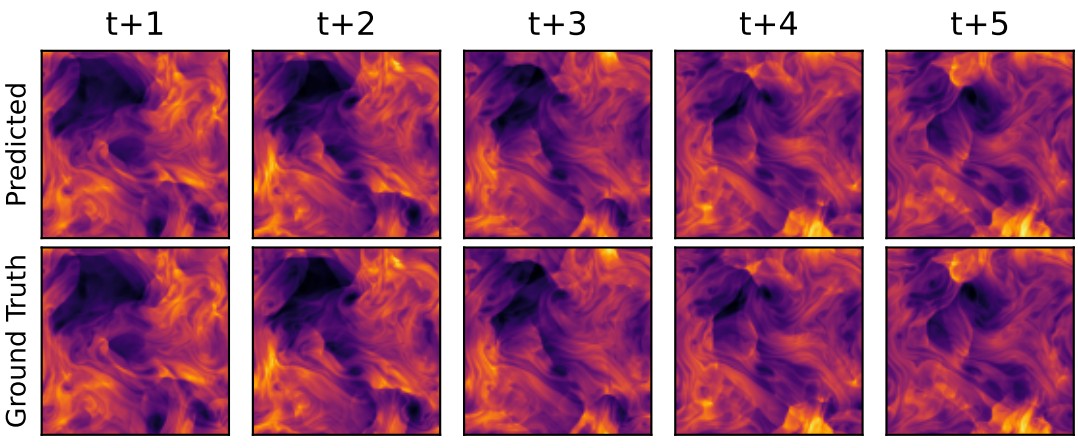

Figure 13: Finetuning trajectory.

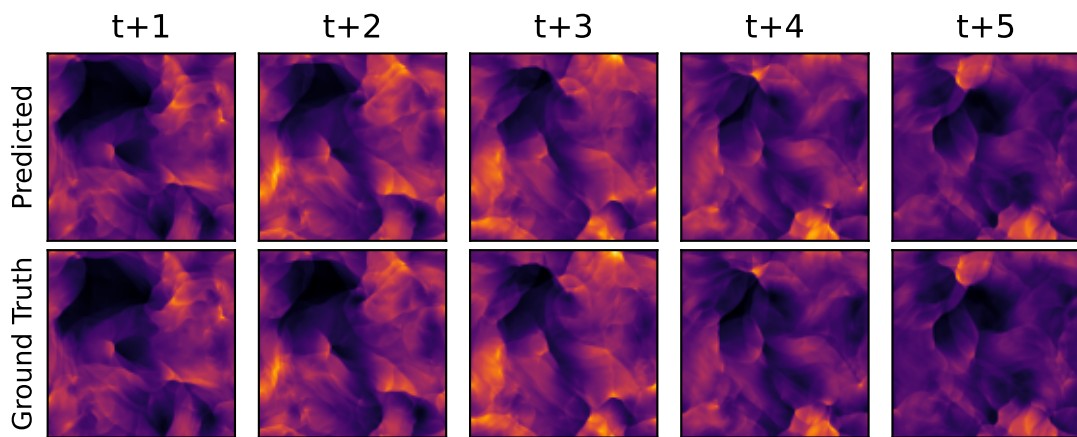

Figure 14: Finetuning trajectory.

