# OpenReview forum: "Multiple Physics Pretraining for Physical Surrogate Models"
_NeurIPS.cc/2023/Workshop/AI4Science — NeurIPS2023-AI4Science Oral_

### Official Review · Reviewer_P7YX · 2023-10-15
**Great Idea combining different physics into a single model**

**Rating:** 7
**Confidence:** 5

**Review:**

This work presents a great idea of combining different physics into a single large model. Results demonstrate outperformance compared to other methods.

To put this into a publication beyond this workshop, it might be worth comparing the effectiveness of your pretrained model with SOTA models in PDE timestepping tasks beyond PDEbench.

---

### Official Review · Reviewer_Zd8x · 2023-10-25
**Multiple Physics Pretraining for Physical Surrogate Models**

**Rating:** 6
**Confidence:** 3

**Review:**

**Summary:**
This paper proposes multiple physics pretraining (MPP) for physical surrogate modeling. MPP trains large surrogate models to predict the dynamics of multiple heterogeneous physical systems at the same time by projecting the fields of multiple systems into shared embedding space. They validate the efficacy of MPP on both pretraining and downstream tasks.

**Strengths and Weakness:**
- Overall this paper is well-written, and the presentation is clear and easy to follow.
- The pretrained transformer model can match or surpass modern baselines trained only on specific pretraining sub-tasks without finetuning.
- The pretrained model has the potential to be transferred to solve other systems with limited training samples. But in section 5.2 on transfer learning, authors only compare MPP with VideoMAE which is not specifically designed for solving PDEs. No other baseline models are compared in this low-data setting, such as FNO, FFNO [1], GFNO [2].

[1] Tran, Alasdair, et al. "Factorized fourier neural operators." arXiv preprint arXiv:2111.13802 (2021).\
[2]  Helwig, Jacob, et al. "Group Equivariant Fourier Neural Operators for Partial Differential Equations." arXiv preprint arXiv:2306.05697 (2023).


**Limitations**:
The limitations of this paper is well discussed.

---

### Meta-Review · Area_Chair_ZYjs · 2023-10-27

**Recommendation:** Accept (Oral)
**Confidence:** 4

**Metareview:**

In this manuscript, the authors present the novel concept of multiple physics pretraining (MPP) for physical surrogate modeling. While the idea of physics pretraining has been previously explored, this work distinguishes itself as, to my understanding, the inaugural attempt at pretraining with multiple physics models. Given the intricate complexities inherent in real-world physics, I am of the belief that this approach holds more promise than its single-physics counterpart. Furthermore, the authors commendably benchmark their model against other state-of-the-art deep learning methodologies, such as videoMAE. I hope authors can carefully address the concern from Reviewer Zd8x.